# Sarcopenic obesity in predialysis chronic kidney disease: Muscle strength is a predictor of mortality and disease progression in a six-year prospective cohort

**Dílmerson Oliveira**[1,2©], **Viviane Angelina de Souza**[1*©], **Guilherme Cardenaz Souza**[3], **Lucas Fernandes Suassuna**[1], **Marcus Gomes Bastos**[1], **Maycon Moura Reboredo**[1], **Natália Maria da Silva Fernandes**[1©]

**1** Department of Clinical Medicine, School of Medicine, Federal University of Juiz de Fora, Juiz de Fora, Brazil, **2** Department of physiology, School of Physical Education, FAMINAS, Muriaé, Brazil, **3** Bone D Consultoria e Treinamento, São Paulo, SP, Brazil

© These authors contributed equally to this work.

\* vivi.reumato@gmail.com

## Abstract

### Introduction

Sarcopenic obesity (SO) is a poorly defined physiological condition that is associated with increased morbimortality in the general population. The prevalence of SO has increased, making it an important public health issue; however, its effects when associated with other chronic conditions are still unclear. Therefore, it is necessary to comprehend the potential outcomes in chronic kidney disease (CKD) patients.

### Objective

To assess the associations of predialysis CKD patients with SO and its components with death and disease progression to renal replacement therapy (RRT).

### Methods

Prospective six-year cohort with 100 patients with predialysis CKD (stages one through five). Participants were older than 18 years and signed an informed consent form. The data were collected in an outpatient clinic specializing in predialysis care, and demographic, clinical, laboratory and inflammatory variable data were collected. A descriptive analysis was performed, and the SO and non-SO groups were compared with Student's t test, Mann–Whitney U test and Cox regression, taking into consideration all relevant confounding variables. Body composition variables utilized to diagnose SO were separately analyzed along with the outcomes.

### Results

Sixteen percent of our sample were diagnosed with SO, but this was not associated with death or RRT, while lower BMI values were negatively associated with these outcomes.

**Data availability statement:** All relevant data are within the manuscript and its Supporting Information files.

**Funding:** The author(s) received no specific funding for this work.

**Competing interests:** No authors have competing interests.

However, in an isolated analysis, handgrip strength was correlated with both RRT (p = 0.029) and death (p = 0.003). We recommend that health professionals pay greater attention to muscle strength, striving for earlier assessment, in an effort to better counsel patients and implement actions to increase or preserve strength through nonpharmacological treatment, such as resistance training.

## Introduction

Sarcopenia is a process closely associated with aging and is characterized by a progressive reduction in both muscle quality and quantity. Sarcopenia is usually associated with impairments in locomotion, increased risk of falls and fractures, difficulty performing activities of daily living (ADLs), and increased risk of mortality [1,2]; it has a multifactorial etiology and includes environmental factors, inflammation, physical inactivity, mitochondrial abnormalities, loss of neuromuscular junctions, a decrease in the number of myosatellite cells and hormonal changes [3].

In chronic kidney disease (CKD), similar to that in the general population, the prevalence of sarcopenia varies depending on the utilized protocol. However, a cross-sectional study carried out by our group with 100 predialysis CKD patients revealed a prevalence of 28.7% for sarcopenia, which was more prevalent in the more advanced stages of the disease. This condition is also associated with worse performance in activities of daily living, higher body mass index (BMI), slowed gait speed, decreased functional capacity, and greater prevalence of physical inactivity [4].

Obesity is a multifactorial chronic disease with a complex etiology and is directly associated with increased morbidity and mortality [5]. The prevalence of obesity is growing, representing a challenge for public health worldwide [5]. Sarcopenia, in addition to obesity, has increased in prevalence recently, and the combination of both conditions has been defined as sarcopenic obesity (SO) [1]. SO has been shown to increase morbidity and mortality not only in patients with other clinical conditions but also in the general population, especially when diagnosed in older adults [6,7].

Sarcopenia can be diagnosed by several different diagnostic protocols and is difficult to characterize. Similarly, sarcopenic obesity presents a greater problem since there is no consensus in the literature about its diagnostic criteria and assessment instruments. Precise data regarding the prevalence of SO in the general population and among people with chronic illnesses are also lacking [8–11]. The main findings regarding SO in patients with CKD are a direct relationship with albuminuria, even in patients with normal BMI. Moreover, albuminuria is directly associated with reduced muscle mass in patients with type 2 diabetes [12–16].

Therefore, the aim of the present study was to more thoroughly evaluate potential unfavorable outcomes in CKD patients concomitantly diagnosed with SO.

### Objective

To assess the association between sarcopenic obesity and its components and mortality and disease progression to renal replacement therapy (RRT) in patients with predialysis CKD.

## Methods

### Participants

A prospective cohort study lasting from 2014 to 2022 included 100 patients from the outpatient clinic of the Center for Hypertension and Diabetes (HIPERDIA), a secondary health

center specializing in treating patients with hypertension and cardiovascular risk, diabetes mellitus (DM) I and II with difficult metabolic control, and patients with predialysis CKD. This research was approved by the ethics committee of the University Hospital of the Federal University of Juiz de Fora. All patients signed an informed consent before admission to the study.

The first phase of the study consisted of a cross-sectional evaluation of the patients carried out between 2014 and 2016 and was published as the original article *"Sarcopenia in patients with chronic kidney disease not yet on dialysis: Analysis of the prevalence and associated factors"* [4]. In the second phase of our study, the same patients from the first phase were enrolled and there was no loss of follow up.

The diagnosis of CKD was confirmed and classified by specialists in accordance with Kidney Disease: Improving Global Outcomes (KDIGO) [17]. The glomerular filtration rate was estimated through the CKD Epidemiology Collaboration (CKD-EPI) equation [18]. Demographic data, comorbidity data (classified with ICD-10), and laboratory data were included in the study.

## Measurements and evaluations

Muscle mass was assessed with a GE Lunar Prodigy Primo® densitometer. Appendicular lean mass (ALM) was utilized to determine low muscle mass following two criteria established by consensus. According to Baumgartner and colleagues [19], low muscle mass was considered for values below $7.260 \ kg/m^2$ and $5.545 \ kg/m^2$ for men and women, respectively. In accordance with the Foundation for the National Institute of Health (FNIH) criteria, ALM was divided by BMI, and low muscle mass was defined by values below $0.789 \ kg/m^2$ or $0.512 \ kg/m^2$ in men and women, respectively [20].

Handgrip strength was used to assess muscle strength using a Jamar Hydraulic Hand Dynamometer® with a scale ranging from 0 to 90 kg and a resolution of 2 kg. Three measurements were taken from the dominant limb, while the patient was sitting on a chair, with the elbow flexed in 90º, with a three minute interval between each measurement, the highest measurement was utilized for our analysis. Low muscle strength was determined by values less than 30 kg-f (Kg-f is the measure of force applied by human movement to a dynamometer) and 20 kg-f for men and women, respectively. In accordance with the FNIH criteria, low muscle strength was considered for values below 26 kg-f or 16 kg-f for men and women, respectively. To assess muscle performance, a 6-meter gait speed test was utilized, and values less than 0.8 m/s were considered poor performance. The gait test was performed in an ample and well lit corridor, a measuring tape determined the length of the track plus 2 meters before the starting point and 2 meters after the finish, in order to discount the patient's acceleration and deceleration. Individuals were instructed to walk at their usual pace, and time was measured by research staff (Table 1).

Obesity was defined as a BMI above $27.1 \ kg/m^2$, following criteria established by the World Health Organization (WHO). BMI was stratified by age [21]. Obesity was also assessed by the fat mass index (FMI), and values above 6 for men and 9 for women were considered altered [22]. Finally, body fat percentage (BFP) and body lean mass percentage (BLMP) were also assessed following the criteria of Sharma, Hawkins and Abramowitz [23], establishing specific cutoff points for obesity in predialysis CKD patients.

When assessing SO, the FNIH criteria were utilized in combination with the three aforementioned variables (BMI, FMI and BFP). Based on these data points, the sample was dichotomized into SO and non-SO groups. Regarding body fat, in case BMI and FMI or BFP reached discordant values, FMI and BFP measurements were prioritized (19,20).

**Table 1. Foundation for the National Institutes of Health (FNIH) recommended criteria for sarcopenia.**

| Criteria | Measure | Cut-Point | |
|---|---|---|---|
| | | **Men** | **Women** |
| FNIH | | | |
| Muscle Mass | ALM divided by H² | < 0,789 | < 0,512 |
| Muscle Strength | Handgrip strength | < 30 kg | < 16 kg |
| Muscle Performance | Walking speed | < 0,8 m/s | <0,8 m/s |

FNIH: Foundation for the National Institutes of Health; ALM: appendicular lean mass; BMI: body mass index; H2: squared height.

Blood samples were collected upon admission to the study, and complete blood count; albumin; blood sugar; blood cholesterol; triglyceride; vitamin D; parathormone; calcium; phosphate; and creatinine were evaluated. The levels of high-sensitivity C-reactive protein (hsCRP) and interleukins 6 and 17 (IL-6 and IL-17) were utilized as inflammation markers. Interleukin-4 (IL-4) was utilized as an anti-inflammatory marker. The levels of hsCRP were measured using immunoturbidimetry, and interleukins were measured with an enzyme-linked immunosorbent assay (ELISA) kit. The urine protein/creatinine ratio was determined from a patient's urine sample, and values above 0.20 mg/g were abnormal.

## Outcomes

The following outcomes were considered in the statistical analysis: mortality, disease progression to RRT and remaining in predialysis care. An active search was performed by a trained researcher for all patients evaluated during the first phase of the research. Contact was established via telephone calls, or when possible, mail was sent to the patient's registered address or e-mail. A survey was also performed on electronic medical records and in the national online obituary via the online address http://www.falecidosnobrasil.org.br to evaluate patient outcomes.

## Statistical analysis

The data were analyzed for normality with the Kolmogorov–Smirnov test. Continuous variables are summarized as mean ± standard deviation, and categorical variables are expressed as the number of participants (percentage). The SO and non-SO groups were compared in relation to demographic, laboratory, clinical and outcome data using Student's t test, the Mann–Whitney U test or ANOVA according to the variables' characteristics (ordinal, categorical, numeric, continuous). We performed a Pearson correlation between handgrip strength and BMI. A Cox regression model was utilized to assess the outcomes of death and RRT, adjusting for confounders (age, sex, eGFR, BMI, BFP, BLMP, handgrip strength, and DM).

The software SPSS 25.0 was utilized for the statistical analysis, for which the confidence interval (CI) was 95% and the p value was < 0.05.

## Results

A total of 100 patients participated in our study. The mean age was 73.59 ± 9.22 years, and 59% were female. All patients had hypertension. For the CKD patients, most had stage 3b (35%) or 4 (36%) disease. Out of the total of the sample, 28% had sarcopenia, 36% were diagnosed with obesity and 16% had SO.

Obesity was present in 36%, 28% of the sample had sarcopenia, and 16% had SO (Table 2).

When the samples were divided into SO (n = 16) and non-SO (n = 84) groups, it was possible to observe that both groups were similar in all variables (Table 3).

**Table 2. Demographic, clinical, laboratory and body composition characteristics of the sample.**

|  | n = 100 |
|---|---|
| Age, years (mean ± SD) | 73.59 ± 9.22 |
| Women | 59 (59%) |
| Race-Ethnicity |  |
| White | 29 (29%) |
| Black | 25 (25%) |
| Brown | 46 (46%) |
| Education |  |
| Illiterate | 25 (25%) |
| Incomplete Elementary School | 57 (57%) |
| Complete Elementary School | 5 (5%) |
| Incomplete High School | 2 (2%) |
| Complete High School | 8 (8%) |
| Incomplete Higher Education | 1 (1%) |
| Complete Higher Education | 1 (1%) |
| Marital Status |  |
| Married | 46 (46%) |
| Single | 6 (6%) |
| Divorced | 7 (7%) |
| Widowed | 41 (41%) |
| CKD (stage) |  |
| 1 | 1 (1%) |
| 2 | 6 (6%) |
| 3a | 13 (13%) |
| 3b | 35 (35%) |
| 4 | 36 (36%) |
| 5 | 9 (9%) |
| Alcoholism | 23 (23%) |
| Smoking | 5 (5%) |
| Sarcopenia (FNIH) | 28 (28%) |
| Obesity | 36 (36%) |
| Sarcopenic Obesity (FNIH) | 16 (16%) |
| Outcomes |  |
| Predialysis care | 50 (50%) |
| Renal Replacement Therapy | 11 (11%) |
| Death | 39 (39%) |
| Mean of follow-up, months (mean ± SD) | 71.99 ± 22.16 |
| Glomerular Filtration Rate, mL/min/1,73 m² (mean ± SD) | 44.90 ± 72.26 |
| Smoking, years (mean ± SD) | 16.35 ± 20.77 |
| Systolic Blood Pressure, mmHg (mean ± SD) | 152.00 ± 25.74 |
| Diastolic Blood Pressure, mmHg (mean ± SD) | 87.20 ± 13.49 |
| Body Mass Index, kg/m² (mean ± SD) | 28.75 ± 5.54 |
| Fat Percentage, % (mean ± SD) | 36.50 ± 9.15 |
| Total Body Fat, kg (mean ± SD) | 25.15 ± 9.11 |
| Total Body Lean Mass, kg (mean ± SD) | 41.85 ± 8.68 |
| Abdominal circumference | 92.07 ± 10.05 |
| Creatinine, mg/dL (mean ± SD) | 2.13 ± 0.96 |

*(Continued)*

**Table 2.** (Continued)

|  | n = 100 |
|---|---|
| Protein/Creatinine Ratio, mg/mg (mean ± SD) | 1.92 ± 11.94 |
| Total cholesterol, mg/dL (mean ± SD) | 172.56 ± 41.50 |
| HDL, mg/dL (mean ± SD) | 42.16 ± 10.45 |
| LDL, mg/dL (mean ± SD) | 98.83 ± 35.12 |
| VLDL, mg/dL (mean ± SD) | 31.45 ± 18.72 |
| Triglycerides, mg/dL (mean ± SD) | 145.53 ± 70.12 |
| Glucose, mg/dL (mean ± SD) | 116.54 ± 45.69 |
| Calcium, mg/dL (mean ± SD) | 9.01 ± 0.64 |
| Phosphate, mg/dL (mean ± SD) | 3.93 ± 1.07 |
| Albumin, g/dL (mean ± SD) | 4.01 ± 0.32 |
| Hemoglobin, g/dL (mean ± SD) | 37.99 ± 5.12 |
| PTH, pg/ml (mean ± SD) | 202.56 ± 150.66 |
| Vitamin D, ng/dL (mean ± SD) | 29.46 ± 10.65 |
| hsCRP, mg/dL (mean ± SD) | 8.03 ± 1.20 |
| IL-4, pg/mL (mean ± SD) | 1308.25 ± 561.92 |
| IL-6, pg/mL (mean ± SD) | 642.49 ± 196.01 |
| IL-17, pg/mL (mean ± SD) | 1398.90 ± 636.26 |

Data expressed in mean ± standard deviation, median (minimum and maximum) or frequency (percentage), CKD: Chronic Kidney Disease, FNIH: Foundation for the National Institutes of Health, HDL: High density lipoprotein, LDL: Low density lipoprotein, VLDL: Very Low Density Lipoprotein, PTH: Parathormone, hsCRP: high sensitive C-reactive protein, IL-4: Interleukin-4, IL-6: Interleukin-6, IL-17: Interleukin-17.

There was no statistically significant difference in inflammatory markers between the SO and non-SO groups; however, it is still possible to observe a worse metabolic profile in SO patients, with higher glucose levels (Table 4).

Fig 1 shows that a greater percentage of SO patients had very similar results in inflammatory marker levels, probably due to the small sample size (Fig 1).

ACox regression using RRT as the outcome and adopting BMI as a categorical variable showed that a higher eGFR was associated with a lower risk of RRT (p = 0.015) (Table 5).

Table 6 shows that a normal BMI (p = 0,033) and a lower BMI (p = 0,001) were associated with death (Table 6).

Two other Cox regressions were performed using handgrip strength as a continuous variable and having RRT and death as outcomes, with the same independent variables used in the previous analyses. Handgrip strength significantly protected against disease progression to RRT (p = 0.029) (Table 7) and death (p = 0.003) (Table 8).

According to the final Cox regression model, which included RRT (Table 9) and death (Table 10) as outcomes, with a confidence interval of 95% and SO as an independent variable, it did not significantly influence any of the outcomes (p = 0.888 for RRT and p = 0.925 for death); moreover, the higher the eGFR was at the beginning of the study, the lower the probability of RRT was (p = 0,012).

Abdominal circumference was utilized as a variable that might influence death and RRT outcomes in our analysis; however, this variable was not significantly related to survival.

## Discussion

According to our literature review, the present study is the first to analyze SO and its composing variables according to FNIH criteria in relation to the outcomes of death and disease

**Table 3. Demographic and clinical characteristics of the sample, dichotomized by Sarcopenic Obesity and Non Sarcopenic Obesity (SO).**

|  | Sarcopenic Obesity n (%) | Non-SO n (%) | P value |
|---|---|---|---|
| Total | 16 | 84 |  |
| Sex |  |  |  |
| Men | 6 (14.6) | 35 (85.4) | 0.756 |
| Women | 10 (16.9) | 49 (83.1) |  |
| Race-Ethnicity, n (%) |  |  |  |
| White | 7 (43.7) | 22 (26.2) | 0.312 |
| Black | 4 (25.0) | 21 (25.0) |  |
| Brown | 5 (31.3) | 41 (48.8) |  |
| Education |  |  |  |
| Illiterate | 4 (25.0) | 21 (25.3) | 0.710 |
| Incomplete Elementary School | 10 (62.5) | 47 (56.6) |  |
| Complete Elementary School | 1 (6.3) | 4 (4.8) |  |
| Incomplete High School | 1 (6.3) | 1 (1.2) |  |
| Complete High School | 0 (0%) | 8 (9.6) |  |
| Incomplete Higher Education | 0 (0%) | 1 (1.2) |  |
| Complete Higher Education | 0 (0%) | 1 (1.2) |  |
| Marital Status |  |  |  |
| Married | 2 (12.5) | 44 (52.4) | 0.022 |
| Single | 2 (12.5) | 4 (4.8) |  |
| Divorced | 11 (68.8) | 30 (35.7) |  |
| Widowed | 1 (6.3) | 6 (7.1) |  |
| CKD (stage) |  |  |  |
| 1 | 0 (0.0) | 1 (1.2) | 0.161 |
| 2 | 2 (12.5) | 4 (4.8) |  |
| 3a | 3 (18.8) | 10 (11.9) |  |
| 3b | 7 (43.8) | 28 (33.3) |  |
| 4 | 3 (18.8) | 33 (39.3) |  |
| 5 | 1 (6.3) | 8 (9.5) |  |
| High Blood Pressure | 28 (28%) | 72 (72%) |  |
| Diabetes Mellitus | 18 (64.3) | 36 (50%) | 0.144 |
| Alcoholism | 6 (21.4) | 17 (23.6) | 0.521 |
| Smoking | 1 (3.6) | 4 (5.6) | 0.919 |
| Fat Percentage, % (mean ± SD) | 43.7(8.1) | 35.1(8.6) | <0.0001 |
| Total Body Fat, kg (mean ± SD) Total | 34.1(9.5) | 23.4(7.9) | <0.0001 |
| Abdominal Circumference (cm) | 101.5(9.3) | 90.2(10.1) | <0.0001 |
| Outcomes |  |  | 0.776 |
| Predialysis care | 7 (17.1) | 34 (82.9) |  |
| Renal Replacement Therapy | 1 (9.1) | 10 (90.9) |  |
| Death | 7 (17.9) | 32 (82.1) |  |

Data expressed in mean ± standard deviation, or frequency (percentage). CKD: Chronic Kidney Disease,

*: statistical significance.

progression to RRT in patients with predialysis CKD. This was a prospective cohort study, with an average follow-up time of 71.99 ± 22.16 months, carried out after a previous cross-sectional study [4]. In our sample, 36% had obesity, 28% had sarcopenia, and 16% of our sample had SO

**Table 4. Clinical and laboratory variables dichotomized by Sarcopenic Obesity.**

|  | Non-SO | Sarcopenic obesity | *P* value |
|---|---|---|---|
| Age (years) | 73.44 ± 9.17 | 74.04 ± 9.54 | 0.780 |
| Smoking (years) | 15.93 ± 19.60 | 17.60 ± 24.35 | 0.730 |
| eGFR (ml/min/1.73m$^2$) | 31.55 ± 78.63 | 33.65 ± 9.954 | 0.535 |
| Creatinine (mg/dL) | 2,18 ± 1.01 | 1.81 ± 0.56 | 0.156 |
| Total Cholesterol (mg/dL) | 173.33 ± 41.85 | 168.50 ± 4.69 | 0.672 |
| HDL (mg/dL) | 42.79 ± 10.28 | 38.23 ± 10,98 | 0.145 |
| LDL (mg/dL) | 100.96 ± 35.60 | 85.53 ± 29.85 | 0.143 |
| VLDL (mg/dL) | 29.79 ± 16.69 | 40.12 ± 25.96 | 0.043* |
| Triglycerides (mg/dL) | 139.73 ± 64.03 | 177.60 ± 93.55 | 0.054 |
| Calcium (mg/dL) | 9.03 ± 0.65 | 8.92 ± 0.64 | 0.545 |
| Phosphate (mg/dL) | 3.99 ± 1.12 | 3.63 ± 0.65 | 0.218 |
| Glucose (mg/dL) | 112.55 ± 44.37 | 137.43 ± 48.25 | 0.045* |
| hsCRP (mg/L) | 8.06 ± 20.44 | 7.69 ± 6.29 | 0.965 |
| Albumin (g/dL) | 4.02 ± 0.32 | 3.97 ± 0.29 | 0.579 |
| Hemoglobin (g) | 12.36 ± 1.84 | 12.57 ± 1.22 | 0.675 |
| Vitamin D | 29.56 ± 10.96 | 28.67 ± 8.14 | 0.836 |
| IL- 4 (pg/mL) | 1301.51 ± 563.69 | 1347.87 ± 587.90 | 0.832 |
| IL- 6 (pg/mL) | 640.89 ± 204.21 | 651.86 ± 149.09 | 0.885 |
| IL- 17 (pg/mL) | 1423.85 ± 640.09 | 1252.32 ± 633.84 | 0.486 |

The data are expressed as the mean ± standard deviation. CKD: chronic kidney disease; HDL: high-density lipo-protein; LDL: low-density lipoprotein; VLDL: very low-density lipoprotein; PTH: parathyroid hormone; hsCRP: high-sensitivity C-reactive protein; IL-4: interleukin-4; IL-6: interleukin-6; IL-17: interleukin-17;

*: statistical significance.

according to FNIH criteria. The general prevalence of SO is uncertain in the literature due to a series of factors, especially because of the variability of the criteria utilized [8,24,25].

While not statistically significant, higher levels of IL-6, were found in the SO group than in the non-SO group, which might associate inflammation with the development of SO. The mechanism underlying sarcopenia is unclear, and several theories suggest that, in addition to older age, other factors, such as a reduction in anabolic hormones, an increase in myofibril apoptotic activity, oxidative stress, and an accumulation of proinflammatory cytokines, might play a role [25]. IL-4 has been shown to be relevant for preserving muscle mass and improving insulin sensitivity in mice [26]; however, its effects on muscle tissue energy metabolism remain unclear. The present study showed that, even if not significantly, the SO group had higher values of IL-4, more investigation is needed to better comprehend this marker's role in sarcopenia [26].

A recent review [27] has shown that SO is associated with a higher incidence of DM, incapacity and mortality in certain populations. We did not observe an association between SO and DM in our study; however, the same was not true for the outcome of death, with obesity being a protective factor. This finding corroborates the findings of the Brazilian Peritoneal Dialysis Multicentric Cohort Study (BRAZPD) [28], which also showed a greater risk of mortality in patients with a lower BMI (< 18.5 kg/m$^2$). Among CKD patients on RRT, higher BMI values (normal weight and overweight) have been shown to be protective against negative outcomes and patients in nutritional risk are in need of rigorous assistance [28]. When we approach pre-dialysis patients, there is a lack of studies in the literature, and a study by Kovesdy et al. showed that lower BMI is associated with greater mortality in patients with

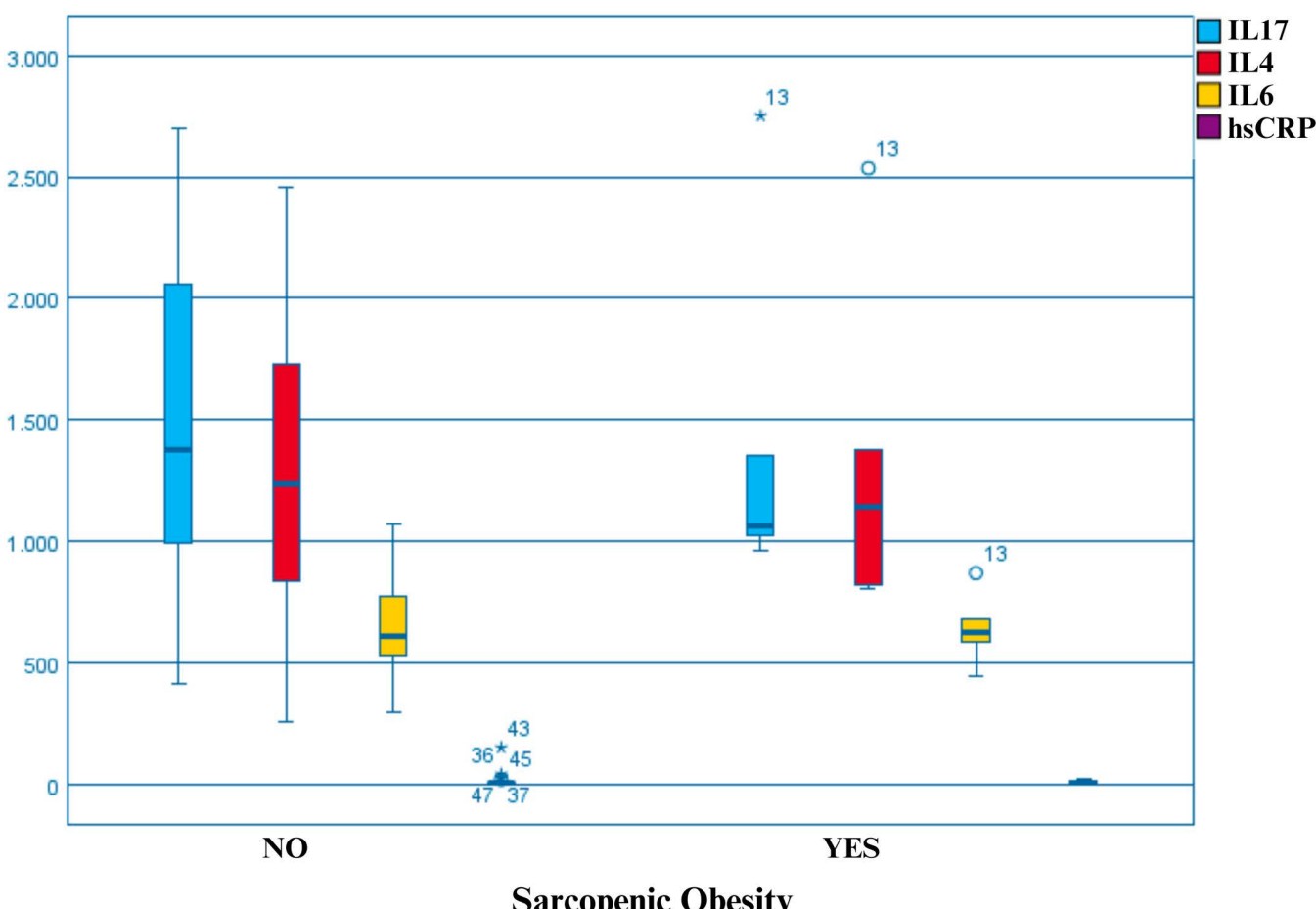

**Fig 1. Inflammatory markers dichotomized by sarcopenic obesity.** IL17 (Interleukin 17); IL4(Interleukin 4); IL6(Interleukin 6), hsCRP (ultrasensitive C-reactive protein), stars and circles represent outliers.

CKD not yet on dialysis therapy. Adjustment for case-mix and surrogate markers of malnutrition and inflammation attenuated, but did not reverse, this relationship. This study was carried out on a sample of 521 men followed for a period of 5.5 years [29]. In contrast, Madero et al. evaluated 1759 individuals in relation to BMI and progression and mortality in CKD and found no association [30]. These studies demonstrate that there is no consensus on this variable and reevaluating it in other contexts is of utmost importance.eGFR was protective against RRT outcome, as expected.

Muscle strength is an important prognostic factor in several chronic conditions. A prospective cohort with 190 clinically stable hemodialysis patients showed that inferior lower limb muscle strength was strongly associated with mortality during a 36-month follow-up [31]. Regarding the handgrip, the first study we found Chang YT et al. evaluated 128 clinically stable patients with non-dialytic CKD and followed up for 33.8 ± 9.2 months demonstrating that handgrip is an independent predictor of composite renal outcomes in non-dialytic CKD patients. This study suggest that handgrip can be incorporated into clinical practice for assessing nutritional status and renal prognosis in patients with non-dialytic CKD [32]. Handgrip strength is suggested as an indirect assessment of nutritional status in CKD patients, but evidence is limited for non-dialysis-dependent CKD patients. As cross-sectional study included 404 patients from the Phase II KoreaN

**Table 5. Cox regression for the outcomes RRT, adopting BMI as a categorical variable.**

|  | HR | CI 95% | P value |
| --- | --- | --- | --- |
| Age | 0.956 | 0.873–1.048 | 0.339 |
| Sex | 1.350 | 0.331–5.501 | 0.676 |
| eGFR | 0.884 | 0.800–0.976 | 0.015* |
| BMI > 25 |  | REFERENCE |  |
| BMI 18,5–24,9 | 1.959 | 0.148–25.925 | 0.610 |
| BMI < 18,5 | 1.649 | 0.379–7.167 | 0.505 |

RRT: Renal Replacement Therapy, HR: hazard ratio, eGFR, estimated Glomerular Filtration Rate, BMI: Body Mass Index,

*: statistical significance.

**Table 6. Cox regression for the outcomes death, adopting BMI as a categorical variable.**

|  | HR | CI 95% | P value |
| --- | --- | --- | --- |
| Age | 1.044 | 1.006–1.084 | 0.023* |
| Sex | 1.023 | 0.534–1.960 | 0.946 |
| eGFR | 1.004 | 0.996–1.012 | 0.298 |
| BMI > 25 |  | REFERENCE |  |
| BMI 18,5–24,9 | 2.842 | 1.087–7.429 | 0.033* |
| BMI < 18,5 | 3.275 | 1.602–6.692 | 0.001* |

HR: hazard ratio, eGFR, estimated Glomerular Filtration Rate, BMI: Body Mass Index,

*: statistical significance.

**Table 7. Cox regression for the outcomes RRT, adopting all body composition variables.**

|  | HR | CI 95% | P value |
| --- | --- | --- | --- |
| Age | 0.983 | 0.849–1.139 | 0.821 |
| Sex | 0.307 | 0.001–95.373 | 0.687 |
| eGFR | 0.879 | 0.777–0.995 | 0.042* |
| BMI | 1.074 | 0.692–1.668 | 0.758 |
| Body Fat percentage | 0.848 | 0.662–1.085 | 0.189 |
| Body Lean Mass percentage | 1.204 | 0.827–1.753 | 0.333 |
| Handgrip Strength | 0.843 | 0.723–0.983 | 0.029* |
| DM | 0.951 | 0.123–7.329 | 0.961 |

HR: Hazard Ratio, CI 95%: 95% Confidence Interval. eGFR: Estimated Glomerular Filtration Rate, BMI: Body Mass Index, DM: Diabetes Mellitus,

*: statistical significance.

Cohort Study for Outcome in Patients With CKD. Higher handgrip was significantly associated with lower malnutrition risk after adjustment (per 1 standard deviation increase, adjusted odds ratio, 0.47 [0.30–0.75]) [33]. A Korean study with 5,859 patients older than 50 years and a follow-up from 2006 to 2014 showed that lower handgrip strength was associated with a greater risk of cardiovascular mortality from all causes, except oncological causes [34].

Myosteatosis is a recent concept that is defined by the infiltration of fat into skeletal muscle tissue. A possible mechanism for the association between obesity and myosteatosis is that adipocytes may become saturated, leading to an increase in ectopic fat storage. In muscle tissue, these ectopic fat storages release proinflammatory cytokines, resulting in local inflammation.

**Table 8. Cox regression for the outcomes death, adopting all body composition variables.**

|  | HR | CI 95% | *P* value |
|---|---|---|---|
| Age | 1.021 | 0.980–1.063 | 0.323 |
| Sex | 2.373 | 0.570–9.875 | 0.235 |
| eGFR | 1.007 | 0.998–1.016 | 0.122 |
| BMI | 0.999 | 0.849–1.176 | 0.992 |
| Body Fat percentage | 0.933 | 0.854–1.019 | 0.123 |
| Body Lean Mass percentage | 1.014 | 0.918–1.120 | 0.782 |
| Handgrip Strength | 0.907 | 0.850–0.968 | 0.003* |
| DM | 0.779 | 0.367–1.651 | 0.514 |

HR: Hazard Ratio, CI 95%: 95% Confidence Interval. eGFR: Estimated Glomerular Filtration Rate, BMI: Body Mass Index, DM: Diabetes Mellitus,

*: statistical significance.

**Table 9. Cox regression for the outcomes RRT, adopting Sarcopenic Obesity.**

|  | HR | CI 95% | *P* value |
|---|---|---|---|
| Age | 0.942 | 0.862–1.029 | 0.186 |
| Sex | 1.413 | 0.351–5.689 | 0.626 |
| eGFR | 0.887 | 0.808–0.974 | 0.012* |
| SO | 0.849 | 0.087–8.251 | 0.888 |

HR: Hazard Ratio, CI 95%: 95% Confidence Interval. eGFR: Estimated Glomerular Filtration Rate, SO: Sarcopenic Obesity,

*: statistical significance.

**Table 10. Cox regression for the outcomes death, adopting Sarcopenic Obesity.**

|  | HR | CI 95% | *P* value |
|---|---|---|---|
| Age | 1.007 | 0.971–1.044 | 0.701 |
| Sex | 0,943 | 0.494–1.799 | 0.858 |
| eGFR | 1.012 | 0.985–1.040 | 0,386 |
| SO | 0,961 | 0.418–2.210 | 0.925 |

HR: Hazard Ratio, CI 95%: 95% Confidence Interval. eGFR: Estimated Glomerular Filtration Rate, SO: Sarcopenic Obesity,

*: statistical significance.

This adipose muscular infiltration occurs in great proximity to muscle fibers; therefore, myosteatosis and obesity may amplify each other, leading to greater loss of muscle function, this may contribute to reducing hand grip strength [35]. Handgrip strength, when analyzed via Cox regression as a continuous variable in relation to both death and RRT outcomes, was the only sarcopenia component with statistical significance in our study; this might indicate that the quality of muscle mass rather than the quantity of muscle mass is more useful as a predictor of poor outcomes [35].

About non-pharmacological interventions to improve physical function in adults with end-stage renal disease, one systematic review and network meta-analysis for randomized controlled trials of a total 44 eligible trials enrolled 2,250 participants, and 16 interventions were identified. It was found that combined resistance and aerobic exercise is the most effective intervention with improvement in several parameters including handgrip and fatigue [36].

A relevant characteristic of our study is the prospective duration of six years with no loss to follow-up. Despite the sample size, the number of outcomes allowed for a robust data analysis. The study limitations include the fact that it was carried out at only one medical center, and it was not possible to establish the causes of death. Additionally, after the second year of follow-up, patient status changes, such as physical activity, new therapies, and new comorbidities, were not considered.

## Conclusion

In our study, there was a SO prevalence of 16% in the predialysis CKD population, but it was not associated with RRT or death outcomes. When analyzing the various components of sarcopenia, it was possible to observe a correlation between handgrip strength and both outcomes. These findings show the importance of early assessment of muscle strength in predialysis CKD patients. This approach would allow for the implementation of actions aiming to preserve or increase muscle strength utilizing nonpharmacological strategies, such as resistance training.

## Author contributions

**Conceptualization:** Dílmerson Oliveira, Viviane Angelina de Souza, Marcus Gomes Bastos, Maycon Moura Reboredo, Natália Maria da Silva Fernandes.

**Data curation:** Dílmerson Oliveira, Viviane Angelina de Souza, Natália Maria da Silva Fernandes.

**Formal analysis:** Dílmerson Oliveira, Viviane Angelina de Souza, Guilherme Cardenaz Souza, Natália Maria da Silva Fernandes.

**Investigation:** Dílmerson Oliveira, Viviane Angelina de Souza, Lucas Fernandes Suassuna, Marcus Gomes Bastos, Natália Maria da Silva Fernandes.

**Methodology:** Dílmerson Oliveira, Viviane Angelina de Souza, Marcus Gomes Bastos, Maycon Moura Reboredo, Natália Maria da Silva Fernandes.

**Project administration:** Dílmerson Oliveira, Viviane Angelina de Souza, Natália Maria da Silva Fernandes.

**Resources:** Dílmerson Oliveira, Viviane Angelina de Souza, Guilherme Cardenaz Souza, Lucas Fernandes Suassuna, Marcus Gomes Bastos, Maycon Moura Reboredo, Natália Maria da Silva Fernandes.

**Supervision:** Dílmerson Oliveira, Viviane Angelina de Souza, Natália Maria da Silva Fernandes.

**Visualization:** Dílmerson Oliveira, Viviane Angelina de Souza, Natália Maria da Silva Fernandes.

**Writing – original draft:** Dílmerson Oliveira, Viviane Angelina de Souza, Lucas Fernandes Suassuna, Maycon Moura Reboredo, Natália Maria da Silva Fernandes.

**Writing – review & editing:** Dílmerson Oliveira, Lucas Fernandes Suassuna, Maycon Moura Reboredo, Natália Maria da Silva Fernandes.

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
