## [Decision Letter · Decision Letter 0]

25 Sep 2024

PONE-D-24-04086Sarcopenic obesity in predialysis chronic kidney disease: muscle strength is a predictor of mortality and disease progression in a six-year prospective cohortPLOS ONE

Dear Dr. Souza,

Thank you for submitting your manuscript to PLOS ONE. After careful consideration, we feel that it has merit but does not fully meet PLOS ONE’s publication criteria as it currently stands. Therefore, we invite you to submit a revised version of the manuscript that addresses the points raised during the review process.

 Both reviewers found merit in your manuscript but clarification on several points are needed. Some statements will need to be referenced and please provide the rationale for why certain criteria was selected. This was brought up by one of the reviewers, so please make sure that is addressed not only in the rebuttal but also in the manuscript.

We look forward to receiving your revised manuscript.

Kind regards,

Jeremy P Loenneke

Academic Editor

PLOS ONE

2. Please ensure you have stated the date of participant recruitment for the present study in the Methods section of your manuscript text to fully comply with the PLOS ONE policy on reporting research involving human participants. This information is currently provided only in the Human Participants Research Checklist, which will not be published with your manuscript files.

Reviewers' comments:

Reviewer's Responses to Questions

**Comments to the Author**

1. Is the manuscript technically sound, and do the data support the conclusions?

Reviewer #1: Yes

Reviewer #2: Partly

2. Has the statistical analysis been performed appropriately and rigorously?

Reviewer #1: Yes

Reviewer #2: Yes

3. Have the authors made all data underlying the findings in their manuscript fully available?

Reviewer #1: Yes

Reviewer #2: Yes

4. Is the manuscript presented in an intelligible fashion and written in standard English?

Reviewer #1: Yes

Reviewer #2: Yes

5. Review Comments to the Author

Reviewer #1: Excellent manuscript and fine publication on this complicated topic.

Abstract

Include low and moderate BMI and death results.

Methods

Define kg/f

Discussion

The results from table 6 ie. BMI 18.5-24.9 and BMI < 18.5 should be discussed. These data suggest that low and moderate BMI is a cause for concern as is low grip strength.

Reviewer #2: This study aims to assess the associations of predialysis CKD patients with sarcopenic obesity and its components with death and disease progression to renal replacement therapy. This is an important topic and the information gained has the potential to positively benefit both patients with CKD and clinicians.

However, there are several concerns that need to be addressed by the authors.

Lines 58-59: reference is needed.

Lines 69-71: “The main findings regarding SO in patients with CKD are a direct relationship with albuminuria, even in patients with normal BMI”. I was not able to locate the reference in which SO was directly related to CKD patients with normal BMI. Please include. Also, by definition, individuals with a normal BMI are not obese. Therefore, is it possible to have sarcopenic obesity if you have normal BMI?

Ensure all abbreviations used consistently throughout the manuscript and are defined at first use. For example, FNIH and EWGSOP are not define within the text.

Line 104: decimal places should be consistent

Methods:

No rationale is provided for why both the EWGSOP and FNIH criteria were used. Also, only the FNIH criteria was used to assess SO and for analysis. Please provide clarification as to why EWGSOP was included in the methodology or why it was not used to assess SO and included in the analysis.

Participants were recruited from 2014-2022. The first phase of the study consisted of a cross-sectional evaluation of patients carried out between 2014-2016 which also included 100 participants (lines 90-93). Therefore, it seems no additional participant were recruited from 2017-2022, is this accurate? Please also include how many participants were excluded during the enrollment period and the reasons why such participants did not meet inclusion criteria.

Lines 123-125: “FNIH criteria were used in combination with BMI, FMI, and BFP to assess SO”. It is not clear how SO was diagnosed. What specific variables were used? And if multiple body composition variables were used as indicated, how did this change the prevalence of SO based on the variable used? Also, is there precedent for the method used in this study? If so, please include reference.

Please expand upon methods for how grip strength and gait speed were measured. For example, was grip strength determined in the dominant limb or both limbs. Were participants in the seated or standing position? Where participants instructed to walk at their customary gait speed?

Statistical analysis:

Lines 151-153: “associations were analyzed with Pearson correlation coefficient.” This information does not seem to be presented in the manuscript.

Results:

Lines 162-163: “Obesity was present in 36%, 28% of the sample had sarcopenia, and 16% had SO”. Please consider editing this sentence for clarity.

Table 4: authors indicate triglycerides to be significantly different between groups at a p-value of 0.054. This does not meet the level of significance as proposed in the statistical analyses section (p-value < 0.05).

Figure 1 quality makes it difficult to read and interpret.

Its unclear what added information Tables 5 and 6, please consider removing.

Lines 197-199: please include abdominal circumference values in Table 1 if measured.

Were differences in body composition and FNIH outcomes assess between SO and non-SO groups? This information is not presented in the Results.

Discussion:

Lines 211-212: “Higher levels of IL-6 were found in the SO group than non-SO group”. This statement does not seem to be true given that there were no significant differs between groups in IL-6 according to Table 4 or lines 168-170. Please revise.

Lines 212-216: Have the authors given consideration to the differences between primary (aging) versus secondary (disease-induced) sarcopenia and how this might influence the interpretation of the findings?

Lines 218-220: again, the data do not seem to support this statement. Please revise.

Lines 224-226: “greater risk of mortality in patients with lower BMI”. This statement deserves greater attention and discussion regarding the obesity paradox seems warranted.

Lines 234-244: this paragraph seems to lump together myosteatosis and strength. Although there is the potential for some interaction between these concepts, they are distinct from one another.

What are the clinical implications of the findings of this work? Given the importance of strength, what interventions should be considered? Has resistance exercise been shown to beneficial for preserving muscle health and strength in CKD patients?

6. PLOS authors have the option to publish the peer review history of their article (what does this mean? ). If published, this will include your full peer review and any attached files.

**Do you want your identity to be public for this peer review?** For information about this choice, including consent withdrawal, please see our Privacy Policy .

Reviewer #1: **Yes: ** Charles Paul Lambert, PhD

Reviewer #2: **Yes: ** Jared M. Gollie

---

## [Author Response · Author response to Decision Letter 1]

12 Nov 2024

Dear Editor and Reviewers,

We, the authors of the article "Sarcopenic obesity in predialysis chronic kidney disease: Muscle strength is a predictor of mortality and disease progression in a six-year prospective cohort", are pleased and grateful for all the comments and the possibility of publication.

All the suggestions and comments made by the editor and reviewers were promptly addressed and can be seen in the "revised manuscript with track changes" file and also in the "response to reviewers" file, as recommended.

One of the reviewers also requested that the image be improved, and this was promptly done according to the journal's standards, using the PACE tool.

We are certain that this manuscript has become clearer and better after each of your careful considerations were taken into account.

Sincerely.

---

## [Decision Letter · Decision Letter 1]

16 Dec 2024

PONE-D-24-04086R1Sarcopenic obesity in predialysis chronic kidney disease: Muscle strength is a predictor of mortality and disease progression in a six-year prospective cohortPLOS ONE

Dear Dr. Souza,

Thank you for submitting your manuscript to PLOS ONE. After careful consideration, we feel that it has merit but does not fully meet PLOS ONE’s publication criteria as it currently stands. Therefore, we invite you to submit a revised version of the manuscript that addresses the points raised during the review process.

 The reviewer thought you addressed most of the concerns, however, there are a few that remain (labeling of tables, and a bit more discussion) which is why I rendered the decision of "minor revision".

We look forward to receiving your revised manuscript.

Kind regards,

Jeremy P Loenneke

Academic Editor

PLOS ONE

Reviewers' comments:

Reviewer's Responses to Questions

**Comments to the Author**

1. If the authors have adequately addressed your comments raised in a previous round of review and you feel that this manuscript is now acceptable for publication, you may indicate that here to bypass the “Comments to the Author” section, enter your conflict of interest statement in the “Confidential to Editor” section, and submit your "Accept" recommendation.

Reviewer #1: (No Response)

2. Is the manuscript technically sound, and do the data support the conclusions?

Reviewer #1: Yes

3. Has the statistical analysis been performed appropriately and rigorously?

Reviewer #1: Yes

4. Have the authors made all data underlying the findings in their manuscript fully available?

Reviewer #1: Yes

5. Is the manuscript presented in an intelligible fashion and written in standard English?

Reviewer #1: Yes

6. Review Comments to the Author

Reviewer #1: A well done project examining the effects of Sarcopenic obesity on a number of variables. Of note it appears that the heading for Non SO and Sarcopenic Obesity are switched for this table 3. This is evident when one looks at the significant measurements near the bottom of the table. For a discussion of table 6. The authors should discuss other papers where moderate to low bmi is associated with increased risk or rate of death. Additionally, providing evidence or explanation as to why grip strength predicts time to death is warranted. Things such as physical activity, nutrition, and rest or sleep!

7. PLOS authors have the option to publish the peer review history of their article (what does this mean? ). If published, this will include your full peer review and any attached files.

**Do you want your identity to be public for this peer review?** For information about this choice, including consent withdrawal, please see our Privacy Policy .

Reviewer #1: **Yes: ** Charles Paul Lambert, PhD

---

## [Author Response · Author response to Decision Letter 2]

22 Dec 2024

Dear Reviewer,

We thank you again for your careful reading, which will improve the article. We have detailed our responses below.

Comments to the Author

1. If the authors have adequately addressed your comments raised in a previous round of review and you feel that this manuscript is now acceptable for publication, you may indicate that here to bypass the “Comments to the Author” section, enter your conflict of interest statement in the “Confidential to Editor” section, and submit your "Accept" recommendation.

Reviewer #1: (No Response)

2. Is the manuscript technically sound, and do the data support the conclusions?

Reviewer #1: Yes

Thank you for the review.

3. Has the statistical analysis been performed appropriately and rigorously?

Reviewer #1: Yes

Thank you for the review.

4. Have the authors made all data underlying the findings in their manuscript fully available?

Reviewer #1: Yes

Thank you for the review.

5. Is the manuscript presented in an intelligible fashion and written in standard English?

Reviewer #1: Yes

Thank you for the review.

6. Review Comments to the Author

Reviewer #1: A well done project examining the effects of Sarcopenic obesity on a number of variables. Of note it appears that the heading for Non SO and Sarcopenic Obesity are switched for this table 3. This is evident when one looks at the significant measurements near the bottom of the table. For a discussion of table 6. The authors should discuss other papers where moderate to low bmi is associated with increased risk or rate of death. Additionally, providing evidence or explanation as to why grip strength predicts time to death is warranted. Things such as physical activity, nutrition, and rest or sleep!

1- Regarding Table 3, we did not observe the error pointed out by your review. We have provided more details on the title of the table:

Table 3 – Demographic and clinical characteristics of the sample, dichotomized by Sarcopenic Obesity and Non-Sarcopenic Obesity (SO) and we emphasize that this population with sarcopenic obesity has greater fat mass, the data on lean mass refer specifically to sarcopenia.

Fat Percentage, % (mean ± SD)

Total Body Fat, kg (mean ± SD) Total

Abdominal Circumference (cm)

43.7(8.1)

34.1(9.5)

101.5(9.3)

35.1(8.6)

23.4(7.9)

90.2(10.1)

<0.0001

<0.0001

<0.0001

2- Regarding Table 6, we have added another paragraph on low BMI and mortality in this population. This data has been discussed in the literature for some time and reference 28 already mentions it. We have added the following studies:

When we approach pre-dialysis patients, there is a lack of studies in the literature, and a study by Kovesdy et al showed that lower BMI is associated with greater mortality in patients with CKD not yet on dialysis therapy. Adjustment for case-mix and surrogate markers of malnutrition and inflammation attenuated, but did not reverse, this relationship. This study was carried out on a sample of 521 men followed for a period of 5.5 years. In contrast, Madero et al evaluated 1759 individuals in relation to BMI and progression and mortality in CKD and found no association. These studies demonstrate that there is no consensus on this variable and reevaluating it in other contexts is of utmost importance.

Kovesdy CP, Anderson JE, Kalantar-Zadeh K. Paradoxical association between body mass index and mortality in men with CKD not yet on dialysis. Am J Kidney Dis. 2007 May;49(5):581-91. doi: 10.1053/j.ajkd.2007.02.277. PMID: 17472839

Madero M, Sarnak MJ, Wang X, Sceppa CC, Greene T, Beck GJ, Kusek JW, Collins AJ, Levey AS, Menon V. Body mass index and mortality in CKD. Am J Kidney Dis. 2007 Sep;50(3):404-11. doi: 10.1053/j.ajkd.2007.06.004. PMID: 17720519.

Regarding the handgrip, the first study we found (Chang YT) evaluated 128 clinically stable patients with non-dialytic CKD and followed up for 33.8 ± 9.2 months demonstrating that handgrip is an independent predictor of composite renal outcomes in non-dialytic CKD patients. This study suggest that handgrip can be incorporated into clinical practice for assessing nutritional status and renal prognosis in patients with non-dialytic CKD. Handgrip strength is suggested as an indirect assessment of nutritional status in CKD patients, but evidence is limited for non-dialysis-dependent CKD patients. As cross-sectional study included 404 patients from the Phase II KoreaN Cohort Study for Outcome in Patients With CKD. Higher handgrip was significantly associated with lower malnutrition risk after adjustment (per 1 standard deviation increase, adjusted odds ratio, 0.47 [0.30–0.75]) (33x).(KIM et al)

Chang YT, Wu HL, Guo HR, Cheng YY, Tseng CC, Wang MC, Lin CY, Sung JM. Handgrip strength is an independent predictor of renal outcomes in patients with chronic kidney diseases. Nephrol Dial Transplant. 2011 Nov;26(11):3588-95. doi: 10.1093/ndt/gfr013. Epub 2011 Mar 28. PMID: 21444362.

Kim M, Park YW, Im DW, Jeong Y, Noh HJ, Yang SJ, Kang E, Ryu H, Kim J, Koo JR, Na KR, Seong EY, Oh KH. Association of Handgrip Strength and Nutritional Status in Non-Dialysis-Dependent Chronic Kidney Disease Patients: Results from the KNOW-CKD Study. Nutrients. 2024 Jul 26;16(15):2442. doi: 10.3390/nu16152442. PMID: 39125323; PMCID: PMC11314453.

About non-pharmacological interventions to improve physical function in adults with end-stage renal disease, one systematic review and network meta-analysis for randomized controlled trials of a total 44 eligible trials enrolled 2,250 participants, and 16 interventions were identified. It was found that combined resistance and aerobic exercise is the most effective intervention with improvement in several parameters including handgrip and fatigue (Zhao et al).

Zhao Q, He Y, Wu N, Wang L, Dai J, Wang J, Ma J. Non-Pharmacological Interventions to Improve Physical Function in Patients with End-Stage Renal Disease: A Network Meta-Analysis. Am J Nephrol. 2023;54(1-2):35-41. doi: 10.1159/000530219. Epub 2023 Mar 30. PMID: 36996785.

---

## [Decision Letter · Decision Letter 2]

22 Jan 2025

Sarcopenic obesity in predialysis chronic kidney disease: Muscle strength is a predictor of mortality and disease progression in a six-year prospective cohort

PONE-D-24-04086R2

Dear Dr. Souza,

We’re pleased to inform you that your manuscript has been judged scientifically suitable for publication and will be formally accepted for publication once it meets all outstanding technical requirements.

Kind regards,

Jeremy P Loenneke

Academic Editor

PLOS ONE

Additional Editor Comments (optional):

Reviewers' comments:

Reviewer's Responses to Questions

**Comments to the Author**

1. If the authors have adequately addressed your comments raised in a previous round of review and you feel that this manuscript is now acceptable for publication, you may indicate that here to bypass the “Comments to the Author” section, enter your conflict of interest statement in the “Confidential to Editor” section, and submit your "Accept" recommendation.

Reviewer #1: All comments have been addressed

2. Is the manuscript technically sound, and do the data support the conclusions?

Reviewer #1: Yes

3. Has the statistical analysis been performed appropriately and rigorously?

Reviewer #1: Yes

4. Have the authors made all data underlying the findings in their manuscript fully available?

Reviewer #1: Yes

5. Is the manuscript presented in an intelligible fashion and written in standard English?

Reviewer #1: Yes

6. Review Comments to the Author

Reviewer #1: Well done. Excellent manuscript which advances the Sarcopenic obesity field. The authors should parlay this into major funding as this was an unfunded study. The problem of fat and low muscle mass is primarily a problem of a sedentary lifestyle. Regular resistance training and aerobic training can add muscle mass and reduce weight, respectively.

7. PLOS authors have the option to publish the peer review history of their article (what does this mean? ). If published, this will include your full peer review and any attached files.

**Do you want your identity to be public for this peer review?** For information about this choice, including consent withdrawal, please see our Privacy Policy .

Reviewer #1: **Yes: ** Charles Paul Lambert, PhD

---

## [Editor Report · Acceptance letter]

PONE-D-24-04086R2

PLOS ONE

Dear Dr. Souza,

I'm pleased to inform you that your manuscript has been deemed suitable for publication in PLOS ONE. Congratulations! Your manuscript is now being handed over to our production team.

Kind regards,

on behalf of

Dr. Jeremy P Loenneke

Academic Editor

PLOS ONE